# Characterization of LysBC17, a Lytic Endopeptidase from *Bacillus cereus*

**DOI:** 10.3390/antibiotics8030155

**Published:** 2019-09-19

**Authors:** Steven M. Swift, Irina V. Etobayeva, Kevin P. Reid, Jerel J. Waters, Brian B. Oakley, David M. Donovan, Daniel C. Nelson

**Affiliations:** 1USDA, Agricultural Research Service, BARC, Animal Biosciences and Biotechnology Laboratory, Baltimore Ave., Beltsville, MD 10300, USA; steven.swift.phd@gmail.com (S.M.S.); kreid414@gmail.com (K.P.R.); renovatebio.jerelwaters@gmail.com (J.J.W.); ddonovan0@yahoo.com (D.M.D.); 2ContraFect Corporation, Yonkers, NY 10701, USA; 3Institute for Bioscience and Biotechnology Research, Rockville, MD 20850, USA; irina.v.etobayeva.mil@mail.mil; 4Department of Veterinary Medicine, University of Maryland, College Park, MD 20742, USA; 5RenOVAte Biosciences Inc., Reisterstown, MD 21136, USA; 6College of Veterinary Medicine, Western University of Health Sciences, Pomona, CA 91766, USA; boakley@westernu.edu

**Keywords:** peptidoglycan hydrolase, endopeptidase, endolysin, autolysin, *Bacillus cereus*, *Bacillus anthracis*

## Abstract

*Bacillus cereus*, a Gram-positive bacterium, is an agent of food poisoning. *B. cereus* is closely related to *Bacillus anthracis*, a deadly pathogen for humans, and *Bacillus thuringenesis*, an insect pathogen. Due to the growing prevalence of antibiotic resistance in bacteria, alternative antimicrobials are needed. One such alternative is peptidoglycan hydrolase enzymes, which can lyse Gram-positive bacteria when exposed externally. A bioinformatic search for bacteriolytic enzymes led to the discovery of a gene encoding an endolysin-like endopeptidase, LysBC17, which was then cloned from the genome of *B. cereus* strain Bc17. This gene is also present in the *B. cereus* ATCC 14579 genome. The gene for LysBC17 encodes a protein of 281 amino acids. Recombinant LysBC17 was expressed and purified from *E. coli*. Optimal lytic activity against *B. cereus* occurred between pH 7.0 and 8.0, and in the absence of NaCl. The LysBC17 enzyme had lytic activity against strains of *B. cereus*, *B. anthracis*, and other *Bacillus* species.

## 1. Introduction

*Bacillus cereus* is a Gram-positive, spore-forming, rod-shaped bacterium. It is widely distributed throughout the environment and has been isolated from soil, plants, insects, and mammalian digestive tracts. *B. cereus* is a food poisoning agent capable of causing diarrheal or emetic illness (reviewed in [1,2]). It is also an opportunistic infectious agent capable of causing pneumonia, meningitis, and bacteremia in both immunocompetent and immunosuppressed individuals [3]. Additionally, *B. cereus* is closely related (<3% 16S rRNA sequence difference) to *Bacillus anthracis*, a human/mammal pathogen, and *Bacillus thuringiensis*, an insect pathogen [1].

The increasing prevalence of drug-resistant bacteria has led to efforts to reduce the use of antibiotics, with the goal of slowing the spread of drug-resistance in bacterial populations [4,5,6]. One large application of antibiotics has been in livestock feed, where the presence of antibiotics in the feed promotes the growth of animals (e.g., chickens, pigs, and cows), and reduces the incidence of infection and illness [7]. The use of medically important antibiotics has been banned in Europe in animal feed as growth promoters and limited in the United States to veterinarian-directed therapeutics [8,9,10,11,12]. Alternatives to antibiotics in animal feed are desired for growth promotion and for inhibition of pathogens. Similarly, the emergence of multi-drug resistant bacteria has led to a desire for alternatives to traditional antibiotics in medicine. 

One such alternative includes enzymes, like endolysins, that cause osmolyis of Gram-positive bacteria by degrading the peptidoglycan of the bacterial cell wall [13,14]. Endolysins are peptidoglycan hydrolases encoded by bacterial viruses known as bacteriophages, or “phages”. These enzymes are used by the phage to break open the host cell from the inside to release mature phage particles. Gram-positive bacteria lack an outer membrane, and therefore, the external peptidoglycan is vulnerable to the action of these enzymes when exposed to the outside of the cell. Other peptidoglycan hydrolases, like the bacteriocin lysostaphin, viral associated peptidoglycan hydrolases (VAPGHs), and autolysins, can kill Gram-positive bacteria in this manner as well [15,16].

Endolysins can be isolated from both lytic and temperate phages. Lytic phages infect, reproduce, and lyse their host. In contrast, temperate phages have two life cycles. One is immediate reproduction and host lysis similar to lytic phages. However, for the second life cycle, the phages can instead integrate their genome into the genome of the host bacterium, and remain dormant as prophages [17]. With the continuing development of next generation DNA sequencing, it is relatively easy to sequence bacterial genomes. This allows rapid identification of prophage sequences and their endolysin genes within the bacterial genome, and bypasses the need to isolate intact bacteriophages from the environment that target the bacteria of interest [18,19].

Genome sequence data from a *Bacillus cereus* strain, Bc17, was analyzed for the presence of endolysin-like peptidoglycan hydrolases. In this study, we identified the gene for the LysBC17 endopeptidase, cloned, expressed, and purified the recombinant protein, and characterized its lytic activity against *B. cereus* and other bacteria.

## 2. Results and Discussion

### 2.1. Bioinformatics

The LysBC17 gene (NCBI accession MK205288) was discovered during an analysis of unpublished whole-genome sequencing data from *B. cereus* strain Bc17. Using the NCBI website, a BlastN search of the LysBC17 gene showed a 100% identity match to the BC_2677 locus in the *B. cereus* ATCC 14579 genome (accession AE016877.1). From the BC_2677 locus description, this gene is labeled as an L-alanyl-D-glutamate peptidase (accession AAP09633). In the ATCC 14579 genome, this gene is not in a predicted prophage region [20]; the nearest annotated phage gene is separated by more than 90 kb, suggesting this peptidase gene is not phage-encoded but might instead represent an autolysin. Interestingly, the BC_2677 locus contains several neighboring genes predicted to be associated with antibiotic and metal resistance (Appendix A). Curiously, this peptidase gene is not present across all *B. cereus* strains. A BlastN search of the LysBC17 gene against 147 completed *B. cereus* genomes at NCBI (prok_complete_genomes database, *B. cereus* (taxid:1396)) found only 22 hits with greater than 80% coverage and above 85% identity. 

The LysBC17 protein is predicted to have a Peptidase_M15_4 domain (Pfam designation PF13539) in its N-terminal region. The C-terminal region contains two SH3_3 domains, which are associated with binding to the cell wall of bacteria (Figure 1A) [21]. A design containing an N-terminal catalytic domain (CAT) with a C-terminal cell wall binding domain (CWB) resembles the structure of endolysins of Gram-positive bacteriophages [13]. It is this characteristic that made LysBC17 an interesting target to test for bacteriolytic activity. 

Further analysis of the individual domains, done by BlastP against the Viruses (taxid:10239) data set at NCBI, revealed hits to bacteriophage proteins. The best hit for the LysBC17 CWB was the *Bacillus* phage Waukesha92 endolysin (accession YP_009099314), an N-acetylmuramoyl-L-alanine amidase, with an identity match of 76% over 150 aligned amino acids (Appendix A). Moreover, this was the best hit for the full-length LysBC17, when BlastP was done against the Viruses data set, with 69% identity over 176 amino acids, mostly at the C-terminal CWB region. The best hit for the LysBC17 CAT was an endolysin (accession YP_008433324) from the *Bacillus* phage Wip1, with a 70% identity match over 120 amino acids (Appendix A). These matches suggest that the LysBC17 gene could have been retained in bacterial genomes from a prior temperate phage infection with most of the prophage sequences lost over time. Alternately, this gene could be an autolysin or bacteriocin, but LysBC17 lacks a recognizable signal peptide to facilitate secretion (as determined by SignalP), which makes a role as a bacteriocin less likely. There are *Bacillus* autolysins, LysF8819.1 and LysCER057, which have the same endolysin-like modular design of an N-terminal CAT with C-terminal CWB that have been shown to be capable of lysing *B. cereus* and closely related species like *B. thuringiensis* and *B. weihenstephanensis* [15]. Therefore, LysBC17 could be an endolysin captured from an “ancient” infection and re-purposed as an autolysin.

### 2.2. SDS-PAGE and Zymogram Analysis of LysBC17

The recombinant LysBC17 protein was purified by nickel chromatography and analyzed via SDS-PAGE. The protein appeared at its predicted size of 32.5 kDa (Figure 1B). Recombinant proteins with only the CAT or the CWB protiens were also purified and likewise migrated at their predicted sizes of 15.3 kDa and 19.0 kDa, respectively. All proteins had > 90% purity, including a GFP-CWB fusion protein at 46.9 kDa. LysBC17 was able to digest *B. cereus* cell walls embedded in the zymogram gel, generating a clearing band (Figure 1C). In contrast, neither the CAT or CWB proteins alone, as well as the GFP-CWB fusion, were able to generate a clearing zone in the zymogram gel. This suggests that the CAT protein is not active without the CWB of the full-length protein. The proteins without the CAT were not expected to display activity (from the turbidity reduction assay (TRA) described below), and these zymogram results agree with this expectation.

### 2.3. Characterization of LysBC17 by TRA

The TRA was used to test lytic activity of LysBC17 against whole cells in a suspension. In order to identify the linear range of enzyme activity in the assay, LysBC17 was subject to dose-response testing using two-fold serials dilutions to determine the concentrations at which a 50% drop in dose results in a similar drop in lytic activity. This linear response range was determined to occur between 20 µg/mL and 5 µg/mL for LysBC17 (Figure 2A). All recombinant proteins were tested at a high dose of 80 µg/mL (Figure 2B). While LysBC17 showed substantial activity, the CAT protein showed minimal activity, about 3% of the full-length protein (Figure 2B). As expected, neither the CWB nor GFP-CWB proteins showed activity. In the sensitive spot-lysis assay, only the full-length LysBC17 showed detectable activity at any concentration, but surprisingly, as little as 100 ng showed significant lytic activity in this assay (Figure 2C).

### 2.4. Determination of pH and NaCl Optima for LysBC17

LysBC17 was tested for lytic activity in 40 mM borate-phosphate (BP) buffers covering a pH range of 4.0 to 11.0. The enzyme showed greatest activity at pH 7.0 and pH 8.0, with activity at pH 9.0 reduced to 79% (Figure 3A). At pH 6.0 and pH 10.0, ≤10% activity remained, and at pH 4.0, 5.0, and 11.0 no activity was observed. This is very similar to the pH response seen for other *Bacillus* endolysins. PlyP56, an endopeptidase, and PlyN76, an amidase, both had optimal activity between pH 7.0 and pH 9.0 [22]. Similarly, PlyHSE3, another *B. cereus* endolysin with amidase activity, had optimal activity at pH 8.0 with substantial activity at pH 7.0 and pH 9.0 [23].

The effect of salt concentration on the lytic activity of LysBC17 was also tested. LysBC17 had maximal activity in the absence of NaCl and was reduced to 86% activity at 150 mM NaCl, which is physiological saline (Figure 3B). As salt concentration increased, LysBC17 activity decreased, with activity reduced to 52% at 450 mM NaCl, and to 22% at 900 mM NaCl. Other *Bacillus* endolysins, including PlyP56 and LysB4, have shown similar responses to salt concentration, with maximum activity in the 0–50 mM NaCl range and then a sudden (40% residual activity at 100 mM NaCl for PlyP56) or more gradual (~50% activity at 200 mM NaCl for LysB4) drop in lytic activity as the salt concentration was increased [22,24]. 

### 2.5. Thermostability of LysBC17

The thermostability of LysBC17 was tested in its optimal pH and NaCl conditions. LysBC17 was incubated at the target temperature for 15 min, before assaying residual activity at 22 °C. LysBC17 activity was stable at 22 °C compared to 4 °C. However, after exposure to 40 °C it was reduced to 85% activity, at 60 °C it was reduced to 64% activity, and at 70 °C it was reduced to 15% activity (Figure 4). Exposure to 80 °C was fully inactivating. To combat food poisoning from *B. cereus*, LysBC17 is sufficiently stable for use in food processing which occurs at room temperature or under refrigerated conditions. As human body temperature is 37 °C and endolysins have been shown to be non-toxin in human clinical trials [25], LysBC17 could also be stable enough for therapeutic use. Another possibility for use of LysBC17, or other antibacterial enzymes, would be as an antibiotic replacement in animal feed. During animal feed pellet production, the feed is subject to temperatures as high as 80–95 °C for 20 s to 4 min during steam treatment with additional time at elevated temperatures while cooling down [26,27]. LysBC17 thermostability is not sufficient to survive the heat treatment that is part of animal feed pellet production. It is possible that the thermostability of LysBC17 could be increased by mutagenesis of the LysBC17 gene, using bioinformatically designed mutations, by directed evolution, or by chimeragenesis (reviewed in [28]). Additionally, formulations including additives, like glycerol, mannitol, and sucrose can increase enzyme thermostability [29,30].

### 2.6. Screening the LysBC17 Lytic Host Range

The spot lysis assay was used to screen the activity of LysBC17 against a panel of bacteria. As seen in Figure 2C, *B. cereus* cells embedded in BHI semisolid agar were susceptible to the activity of LysBC17 spotted onto the agar. Distinct clearings were created by the three amounts of LysBC17 tested, 10 µg, 1 µg, and 0.1 µg (Figure 2C). Additional *Bacillus* species, including *B. anthracis*, *B. thuringiensis*, and *B. pumilis* were screened in identical spot lysis assays, as well as representative non-*Bacillus* species (Table 1). LysBC17 showed activity against all five strains of *B. cereus*, two strains of *B. anthracis*, two strains of *B. pumilus*, and the one strain of *B. thuringiensis* tested. In contrast, no lytic activity was noted against *Clostridium perfringens*, *Staphylococcus aureus*, or *Streptococcus uberis*. Taken together, the data suggests that of the genera tested, LysBC17 activity is limited to *Bacillus* species.

### 2.7. Binding of LysBC17 CWB

The predicted CWB is expected to target the endolysin to the cell wall of bacteria. To test this, the LysBC17 CWB was fused to GFP to make GFP-CWB (Figure 1A). This recombinant protein was then used to determine the ability of the CWB to target *B. cereus* and other bacilli by fluorescent microscopy (Figure 5). GFP-CWB was observed to bind to *B. cereus* ATCC 4342, ATCC 11778, and ATCC 14579 (Figure 5A–C, respectively). However, it showed poor binding to *B. cereus* ATCC 13061 (Appendix A) despite the fact that ATCC 13061 was lysed very efficiently by LysBC17 (Table 1). This phenomenon was also seen with other *Bacillus* species. For example, GFP-CWB bound *B. anthracis* Ames 35 (Figure 5E), but not *B. anthracis* UM23 (Figure 5F) despite possessing lytic activity on both strains (Table 1). Likewise, GFP-CWB labeled the surface of *B. pumilus* BJ0050 (Figure 5G) but not *B. pumilus* ATCC 700814 (Figure 5H). Because several of the outlier *Bacillus* strains not bound by GFP-CWB were sensitive to the lytic actions of LysBC17, it is not presently clear if these negative binding results are artifacts due to steric hindrance between GFP and the binding epitope, or whether there are other, non-CWB factors that mediate binding in these instances. Nonetheless, further work would be required to elucidate the potential of the LysBC17 CWB for diagnostic applications.

Where binding was observed, cells were generally outlined by the GFP-CWB, with minor variations. For example, *B. cereus* ATCC 11778 cells showed more pronounced binding at the poles and septum (Figure 5B), and *B. thuringiensis* ATCC 10792 cells showed increased binding at the septum (Figure 5D). The CWBs of at least three other endolysins, PlyP56, PlyN74, and PlyT40, can also bind *B. cereus* and *B. anthracis*, and likewise display more intense binding at septal regions [22]. Similar to LysBC17, the CWBs of these three endolysins contain SH3-type domains for binding to the cell surface. These results suggest that the binding site of LysBC17 is shared between multiple related *Bacillus* species but might be less accessible between two strains of the same species. 

## 3. Materials and Methods 

### 3.1. Bioinformatics

The software tools BlastN and BlastP available from the National Center for Biotechnology Information (NCBI) [31] were first used to identify similar genes and proteins to LysBC17. The Pfam database (version 32.0) [32] at the European Bioinformatics Institute (EBI) was also used to search for conserved protein families and domains. Subsequently, Clustal Omega [33], also located at EBI, was used for multiple sequence alignments and the BoxShade server (version 3.21) at the Swiss Institute of Bioinformatics (SIB) was used to visualize the Clustal Omega alignments. Finally, the SignalP (version 4.1) [34] server at DTU Health Tech, University of Denmark, was used to predict the presence of signal peptides. 

### 3.2. Bacteria

Competent *E. coli* DH5α or BL21(DE3) were purchased from Invitrogen (Carlsbad, CA, USA). *E. coli* transformants were grown in Luri-Bertani (LB broth and agar) supplemented with 150 µg/mL ampicillin. Other bacteria were grown in Brain Heart Infusion (BHI) broth aerobically at 37 °C, except for *C. perfringens*, which was grown anaerobically in sealed glass bottles without agitation or on plates in an anaerobic jar with an AnaeroGen oxygen removal packet (Thermo Fisher Scientific, Waltham, MA, USA). The *B. cereus* Bc17 and *C. perfringens* Cp39 strains were provided by Bruce Seal, Poultry Microbiology Safety Research Unit, Agricultural Research Service, U.S. Department of Agriculture (USDA), Athens, GA. All ATCC strains were from the American Type Culture Collection, Manassas, VA. The *B. anthracis* strains were from the Biodefense and Emerging Infections Research Resources (BEI Resources, under ATCC management). *B. pumilus* strains were provided by John Mayo, Department of Biochemistry and Molecular Biology, University of Georgia, Athens, GA. The *S. uberis* and *S. aureus* strains were from Max J. Paape, USDA, Beltsville, MD, USA.

### 3.3. Cloning of LysBC17 and Derivatives

The LysBC17 gene was amplified by polymerase chain reaction (PCR) from the genomic DNA of *B. cereus* strain Bc17. PCR was done using primers lysF: 5’-ggcacacatATGAAATACCACAATAGAAATGTAAGTAATCTGAATA-3’ and lysR: 5’-gagaggctcgagCTTTACAAACGTAACATATTCACCAGAAAC-3’, with Phusion DNA polymerase and 2× Phusion High-Fidelity PCR Master Mix with HF Buffer (New England Biolabs, Ipswich, MA, USA). PCR cycling steps included 95 °C for 5 min, followed by five cycles of touchdown PCR (95 °C denaturation for 30 s, 70 °C annealing for 30 s (−2 °C/cycle), and 72 °C extension for 40 s. Next, 27 cycles used 95 °C denaturation for 30 s, 55 °C annealing for 30 s, and 72 °C extension for 40 s. For amplicons > 600 bp, the extension time was extended to 60 s. The PCR amplicon (864 bp) for full-length LysBC17 was cloned between the NcoI and XhoI sites of the pET21(a) vector (Novagen, EMD-Millipore, Billerica, MA, USA), which adds coding for the amino acids LEHHHHHH to the end of the gene. The LysBC17 CAT was amplified by PCR using the primer catR: 5’ -gagagactcgaggctGGATCCATATTGCAAATGCGGGCTATCCACA -3’, with the lysF primer, and the 390 bp amplicon was cloned into pET21(a) as described above. The LysBC17 CWB was amplified by PCR using the primer cwbF, 5’ -gagagacatatgggcgtcgacggatccAACTACAAAGGGTATGGAACGGATACTT -3’, with the lysR primer, and the 522 amplicon was cloned into pET21(a) as described above. The GFP fusion, GFP-CWB, was made by PCR amplifying GFP from pCIG-Flag-Sox9-IRES-nls-GFP using the EGFP-F, 5’-gagagacatATGGTGAGCAAGGGCGAG-3’, and EGFP-R, 5’- gagagactcgaggctggatccCTTGTACAGCTCGTCCATGCC -3’, primers. pCIG-Flag-Sox9-IRES-nls-GFP was a gift from Martin Cheung [35] (plasmid # 44281 from Addgene, Watertown, MA, USA). The GFP-CWB-pET15bx vector was constructed from the following ligation: GFP (NdeI + BamHI, 720 base pairs (bp)) + CWB (BamHI + XhoI, 489 bp) + pET11a (XhoI + PstI, 1082 bp) + pET15b (PstI + NdeI, 4624 bp). Ligation reactions and plasmids were transformed into chemically competent *E. coli* according to the manufacturer’s (Invitrogen) instructions. DNA insert sequences for all constructs were confirmed by DNA sequencing through the MacrogenUSA service (Rockville, MD, USA).

### 3.4. Recombinant Protein Expression and Purification

BL21(DE3) transformants were grown in 1 L modified LB (15 g tryptone, 8 g yeast extract, and 5 g NaCl per L, pH adjusted to 7.8 with NaOH) with 150 μg/mL ampicillin at 37 °C with shaking at 225 rpm to mid-log phase (i.e., optical density (OD) or absorbance at 600 nm (A_600_) = 0.4 – 0.6). Protein expression was induced with 1 mM isopropyl-β-D-1-thiogalactopyranoside, and cells were incubated with shaking at 225 rpm at 10 °C for 18–20 h. Protein purification was via NiNTA chromatography per the manufacturer’s (Qiagen) instructions with these modifications. Cells were pelleted by centrifugation at 6000× *g* for 20 min, then resuspended in lysis buffer (50 mM NaH_2_PO_4_, 300 mM NaCl, 10 mM imidazole) supplemented with 30% glycerol and 0.1% polyethyleneimine, and disrupted by sonication (Branson Ultrasonics) for 20 min (25% duty cycle time, 60% output) on ice. The resulting solution was centrifuged at 9236 × *g* for 30 min, and the recombinant protein was purified from the supernatant with an Ni-NTA superflow column (Qiagen) by washing with 40 mL of lysis buffer containing 30% glycerol followed by a 20 mL of wash buffer (50 mM NaH_2_PO_4_, 300 mM NaCl, 20 mM imidazole) containing 30% glycerol, and finally eluted in five 1 mL aliquots of elution buffer (50 mM NaH_2_PO_4_, 300 mM NaCl, 250 mM imidazole, 30% glycerol). Protein concentrations were determined at 280 nm using a NanoDrop spectrophotometer (Thermo Fisher Scientific, Waltham, MA, USA) with the calculated molecular weight and extinction coefficient (calculations via ProtParam program at ExPASy website; http://web.expasy.org/cgi-bin/protparam/protparam).

### 3.5. SDS-PAGE and Zymograms

SDS-PAGE gels were made with a standard 40% acrylamide/bis-acrylamide, 19:1 (5% crosslinker) solution (Bio-Rad, Hercules, CA, USA) and N, N, N′, N′-tetramethylethylenediamine (TEMED) and ammonium persulfate (APS) were used to catalyze the crosslinking reaction. 20% separating gels were routinely used with standard Coomassie staining to visualize the smaller masses of the LysBC17 domains. For zymograms, *B. cereus* strain ATCC 14579 cells, 100 mL, were grown to mid-log phase, OD_600_ = 0.4–0.6, and washed twice in PBS (50 mM NaH_2_PO_4_, 150 mM NaCl, pH 7.5), once in H_2_O and resuspended in approximately 600 µL H_2_O. Resuspended cells were added to 5 mL of 20% separating polyacrylamide gel. This gel, with 5 µg of enzyme/protein sample in each lane, was subjected to SDS-PAGE analysis. The gel was rinsed/soaked in deionized H_2_O for 30 min to remove SDS and allow the proteins to refold, then transferred into PBS + 2% Triton X-100 and incubated for two hours at room temperature to visualize clearing zones.

### 3.6. Plate lysis Assays

Plate lysis assays were as described previously [36,37] with modifications. Two mL of 50-fold concentrated frozen viable mid-log cells were washed three times with 10 mL sterile distilled water (sdH_2_O) and resuspended in 1 mL PBS. Twelve mL of melted 50 °C semisolid BHI agar (37 g/L BHI, 0.7% Bacto-agar) was added to the cells and the mixture was poured into a sterile petri dish. Dishes were incubated at room temperature to solidify and then 10 µL of the purified endolysin (10, 1, and 0.1 mg/mL) or buffer control was spotted onto the plate and allowed to air dry for 20 min. The plates were incubated two hours at 37 °C and then scored for clear zones from cell lysis. 

### 3.7. Turbidity Reduction Assays

TRAs were used to characterize endolysin activity in solution using a modified previously described method [38]. TRAs were performed in a 96-well plate format using frozen viable *Bacillus* cells from mid-log phase cultures aerobically grown at 37 °C. Four mL of 50-fold concentrated cells in PBS, 25% glycerol, pH 7.5 were thawed, washed three times with 10 mL sdH_2_O, and resuspended in sdH_2_O such that a mixture of 0.1 mL cells mixed with 0.1 mL sdH_2_O yielded an OD_600_ nm of 1.0. For TRA experiments, 0.1 mL enzyme was added to each well followed by 0.1 mL bacterial cells and the plate was read in a SpectraMax 340 plate reader (Molecular Devices, Sunnyvale, CA, USA) once every 20 s for 30 min with 3 s shaking between reads at 22 °C. SoftMax Pro software (Molecular Devices, Sunnyvale, CA, USA) was used to collect and analyze the data. To control for settling of bacterial cells over time despite intermittent shaking, controls containing no enzyme were included with each experiment and the baseline values were subtracted from the raw data of the experimental groups. Activity was calculated from the slope (i.e., velocity) of the linear portion of the turbidity reduction curve as measured in loss of optical density per minute. For relative activity, the maximal value within each experiment was set to 100%. Data presented in histograms were analyzed in Microsoft Excel with error bars from the standard deviation function. Each experiment was performed three times with triplicate samples in each experiment. Endolysin dose response curves were done on *B. cereus* strain ATCC 14579. Enzymes were serial diluted ten-fold into SD buffer (50 mM NaH_2_PO_4_, 30 mM NaCl, 25 mM imidazole, 3% glycerol, pH 7.0) to create the dose response. All other TRAs used *B. cereus* strain ATCC 4342. LysBC17 in 50 mM NaH_2_PO_4_ was diluted fifty-fold into the assay buffers used for examining optimal pH and NaCl concentrations, and for thermostability tests. For pH response testing, LysBC17 was put into 40 mM borate-phosphate (BP) buffers, ranging between pH 3 to pH 10. For NaCl response, LysBC17 was put into 50 mM NaH_2_PO_4_, pH 7.0 with NaCl concentrations ranging from zero to 1.8 M NaCl, which was then two-fold diluted by the addition of cells in water, to achieve concentrations from zero to 0.9 M NaCl. For thermostability testing, 10 µg/mL endolysin in 50 mM NaH_2_PO_4_, pH 7.0 was heat-treated at the target temperature for 15 min, then cooled to 4 °C for 10 min using PTC-200 thermocyclers (MJ Research, Bio-Rad, Hercules, CA, USA) prior to testing for residual activity by TRA.

### 3.8. Binding Assays

Overnight cultures of bacteria were pelleted by centrifugation at 5000 rpm for 10 min at 4 °C, resuspended in sterile PBS, and washed in PBS a second time. One hundred µL of cell suspensions were mixed with 10 µL of GFP-CWB protein (5 mg/mL), and incubated on ice for 10 min. After incubation, labeled bacterial cells were pelleted and washed with ice-cold PBS and resuspended in 100 µL PBS again. A sample of this mixture was applied to a glass slide, sealed with a glass coverslip, and images were captured with an Eclipse 80i epifluorescent microscope (Nikon, Tokyo, Japan), using NIS-Elements software (version number 3.22.15, Nikon) for image analysis. The exposure time in the fluorescent channel was 100 ms for all images, so the intensities are directly comparable.

## 4. Conclusions

The gene for the bacteriolytic enzyme LysBC17 was found by bioinformatic searches of bacterial genome sequencing data. This method for lytic enzyme discovery is becoming more prevalent due to the power of next generation sequencing and increasing availability of bacterial genome sequences. LysBC17 was fully active under physiological salt and pH conditions, and was sufficiently thermostable to be considered for human therapeutic applications, but may require protein engineering strategies to survive the elevated temperatures required for animal feed processing applications. LysBC17 lysed every *Bacillus* strain tested as determined by a plate lysis assay, including five *B. cereus* strains, two *B. anthracis* strains, and two *B. pumilus* strains, but did not lyse any of the non-*Bacillus* species tested, suggesting that LysBC17 might have potential as a therapeutic for *Bacillus* infections. In contrast, a GFP fusion to the LysBC17 CWB showed a spectrum of binding efficiency within the group of lysis-sensitive strains, ranging from binding the full-length of the cell wall, to septal-specific binding, to negligible binding, suggesting that realizing any diagnostic potential of the GFP-CWB construct would require further study. 

## Figures and Tables

**Figure 1 antibiotics-08-00155-f001:**
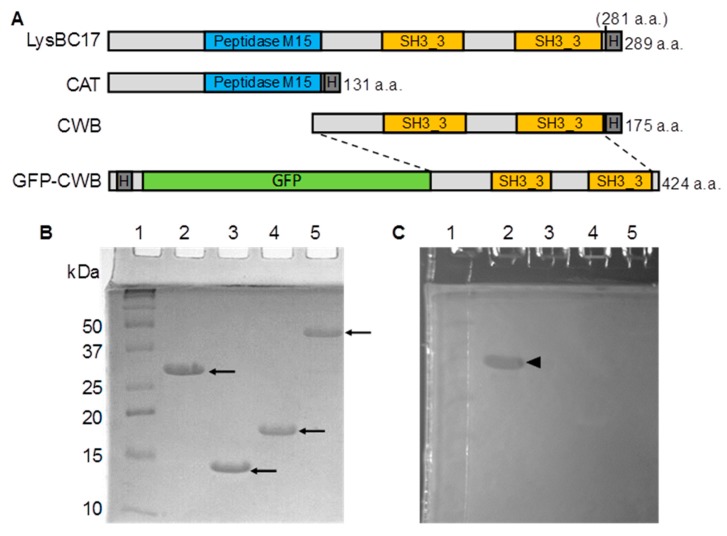
Schematics, sodium dodecyl sulfate polyacrylamide gel electrophoresis (SDS-PAGE), and zymogram of LysBC17, and its recombinant derivatives. (**A**) Schematics showing predicted domains of LysBC17, CAT (catalytic domain), cell wall binding domain (CWB), and the green fluorescent protein (GFP)-CWB proteins. (**B**) Coomassie-stained SDS-PAGE gel. Arrows indicate the purified proteins. (**C**) Zymogram gel with embedded *B. cereus* ATCC 14579 cells. Black triangle marks the clearing from LysBC17 activity on the embedded cells. Lanes for gels: lane 1, markers; lane 2, LysBC17 (32.5 kDa); lane 3, CAT (15.3 kDa); lane 4, CWB (19.0 kDa); lane 5, GFP-CWB (46.9 kDa).

**Figure 2 antibiotics-08-00155-f002:**
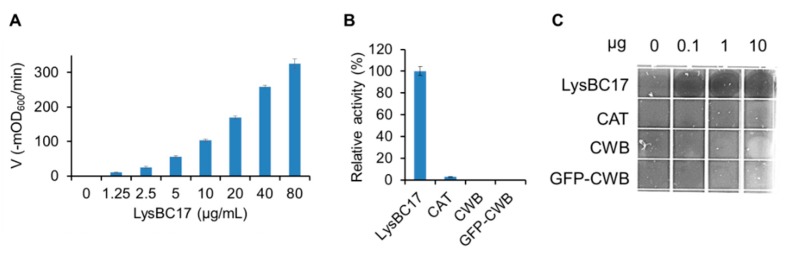
Lytic activity of LysBC17 and its derivatives. (**A**) Dose-response of LysBC17 activity by turbidity reduction assay (TRA). Activity velocity is measured as the drop in milli-optical density at λ600 nm (-mOD_600_) per minute. (**B**) Activity of full-length LysBC17 by TRA compared to its CAT, CWB, and a GFP-CWB fusion; each at 80 µg/mL. Experiments run, *N* = 3. Relative activity of 100% assigned to the maximum value in the data set. (**C**) Spot lysis assay of LysBC17 and its derivatives. *B. cereus* cells were embedded in BHI semisolid agar, and 10 µL of 0.01, 0.1, or 1.0 mg/mL endolysin or buffer control (“0”) were spotted on the top. The plate was incubated at 37 °C for 2 h. Error bars represent the standard deviation.

**Figure 3 antibiotics-08-00155-f003:**
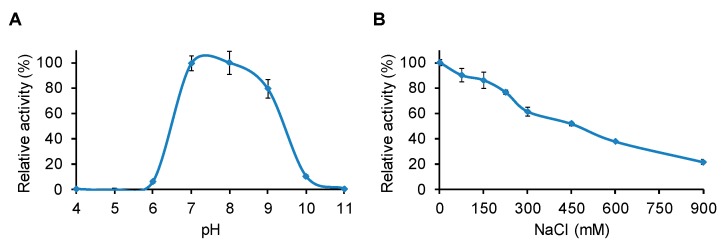
Determination of pH and NaCl optima for LysBC17 by TRA. (**A**) pH range for LysBC17 activity (5 µg/mL) was tested in borate-phosphate buffer versus *B. cereus* cells. (**B**) Range of NaCl for LysBC17 activity (5 µg/mL) was tested in 50 mM NaH_2_PO_4_, pH 7.0 versus *B. cereus* cells. *N* = 3 for all experiments. Relative activity of 100% assigned to the maximum value in the data set. Error bars represent the standard deviation.

**Figure 4 antibiotics-08-00155-f004:**
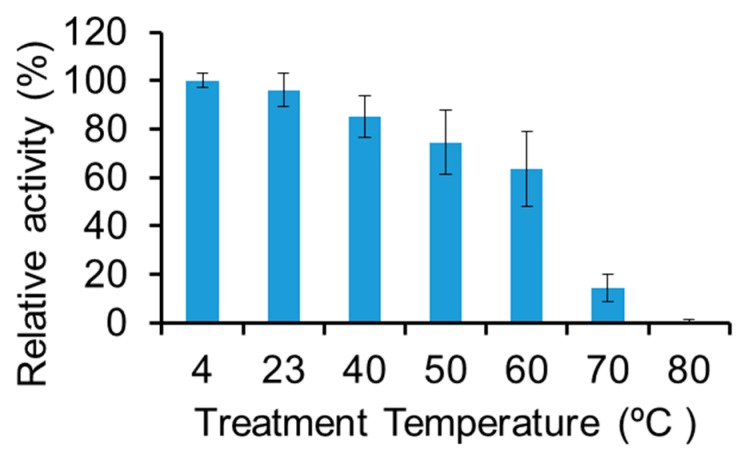
Thermostability of LysBC17. LysBC17 was incubated at the target temperature for 15 min, and then subjected to turbidity reduction assay to determine residual activity. Experiments run, *N* = 3. Relative activity of 100% assigned to the maximum value in the data set. Error bars represent the standard deviation.

**Figure 5 antibiotics-08-00155-f005:**
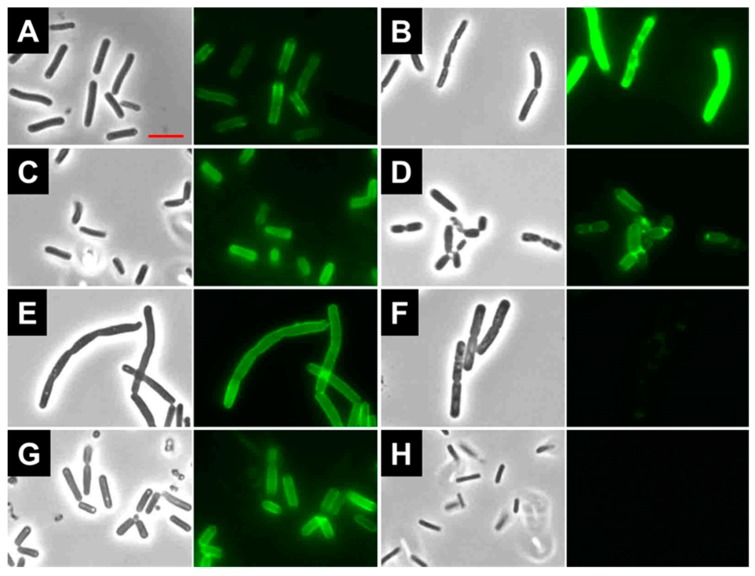
The CWB of LysBC17 directs GFP (GFP-CWB) to the surface of bacteria. Selected bacterial cells and GFP-labeled LysBC17 CWB were mixed together, washed, and viewed by microscopy per the Methods. 1000× phase-contrast images (left panels) are shown with their corresponding fluorescent images (right panels). Exposure time for all fluorescent images was 100 ms. (**A**) *B. cereus* ATCC 4342, (**B**) *B. cereus* ATCC 11778, (**C**) *B. cereus* ATCC 14579, (**D**) *B. thuringiensis* ATCC 10792, (**E**) *B. anthracis* Ames 35, (**F**) *B. anthracis* UM23, (**G**) *B. pumilus* BJ0050, H) *B. pumilus* ATCC 700814. Scale bar = 5 µm.

**Table 1 antibiotics-08-00155-t001:** Bacterial spectrum of LysBC17 lytic activity.

Bacteria	LysBC17 Activity *^a^*
*Bacillus cereus* Bc17	+++
*Bacillus cereus* ATCC 4342	+++
*Bacillus cereus* ATCC 11778	+++
*Bacillus cereus* ATCC 13061	+++
*Bacillus cereus* ATCC 14579	+++
*Bacillus anthracis* Ames 35	++
*Bacillus anthracis* UM23	+
*Bacillus thuringiensis* ATCC 13061	++
*Bacillus pumilus* BJ0050	+++
*Bacillus pumilus* BJ0055	+/-
*Clostridium perfringens* Cp35	-
*Staphylococcus aureus* Newman	-
*Streptococcus uberis*	-

*^a^* Ten µL spots were deposited onto a plate with cells embedded in semisolid agar, and allowed to incubate at 37 °C for 2 h. Plates were scored for clearing zones as follows: +++ = activity at 0.1 µg; ++ = activity at 1 µg; + = activity at 10 µg; +/- = weak activity at 10 µg (i.e., clearing zone, but not completely transparent); - = no activity.

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
