# Peer review of "Characterization of LysBC17, a Lytic Endopeptidase from Bacillus cereus"

_antibiotics, 2019, doi:10.3390/antibiotics8030155_

Round 1
Reviewer 1 Report
The manuscript describes a novel endolysin, LysBC17 from Bacillus cereus strain Bc17. The entire open reading frame that encodes LysBC17 was cloned and sequenced. SDS-PAGE results of LysBC17 in Escherichia coli show that the size of the this enzyme is 32.5 kDa. The optimum pH, NaCl concentration and temperature conditions for LysBC17 activity are determined. LysBc17 activity, however, is limited to Bacillus species.
Table 1 should include Bacillus thuringiensis ATCC 10792 in Figure 5D.
Figure 5 legend needs to provide more details. For instance, I assume these are phase contrast and fluorescent micrographs of GFP-CWB. It does not say what exactly what they are.
Reviewer 2 Report
The manuscript is written well, however, there are gaps that need to be filled (Materials and Methods), figures corrected, genome to be deposited, etc. I recommend major revision.
L34: causing pneumonia?
L34-5: in immunocompromised immunocompromisedpeople?
L49, 55 and throughout: Just a minor remark. English is not my first language, however, "phage" as singular and plural form sounds very unusual to me. Here is an opinion article on this subject: https://www.ncbi.nlm.nih.gov/pmc/articles/PMC3109450/.
L69: analysis or search?
L69-70: Are you planning to deposit the genome sequence? At this point, I have to trust your words, and I cannot verify the data.
L71: NCBI genome/gene access number and NCBI database/site reference in the main text missing.
Figure S1: Please include the length of the region, and rethink the color scheme to optimize the contrast
L77-8: Phylogenetic support would be greatly appreciated.
L81: This suggestion is too strong for provided data, number of fully sequenced genomes. I would avoid it.
Figures S2 and S3: Please increase the resolution
Figure 2a: Control (0 ug/mL) missing. Please invert the x-axis, and explain the y-axis unit/quantity.
Figure 2b: control missing.
Figure 2c: The unit in this panel (ug) does not correspond to the one described in the caption (ug/mL). Please unify. The orientation of the image is counterintuitive (more--> less)
L145: Please define "very little"
L150: optima?
Figure 3: Please justify in the manuscript why these two optima were assessed in two different buffers, and not the same
L159: where was sudded and where gradual? Please explain
L168: Is there any information available on protein's effect on human cells? Please specify
L175: or directed evolution
Figure 4: Please make sure that all figure styles are unified (cropped borders)
All figures: Please define error bars (SD or SEM?)
L185: several=how many?
L191 and throughout: please use non-breaking space between values and units
L193: please define weak
L200: data not shown - please include it in supplemental material
Figure 5: ruler and magnification missing. Analysis with ImageJ or similar software would strengthen the results and make the description much easier in he paragraph above.
Materials and methods
Handling of anaerobes missing (at least brief description/reference) cycling conditions and mastermix composition missing transformation protocol missing L258: how did you adjust the pH? L265: 20 min? That seems a lot. L277: details about the gel missing (acrylamide:bis etc) Bioinformatic part completely missing.
Round 2
Reviewer 2 Report
Thank you for corrections, clarifications, and explanations. I can endorse the manuscript for publication after the authors address the following minor remarks:
L34: producing pneumonia - I was not clear in my previous review report. Please replace the word "producing" with "causing"
L77-78 (phylogenetic support): I had in mind a phylogenetic tree. We can also survive without this.
Throughout: Please cite NCBI, Pfam, and ALL other online tools/software/platforms correctly. Examples:
- Madden T. The BLAST Sequence Analysis Tool. 2002 Oct 9 [Updated 2003 Aug 13]. In: McEntyre J, Ostell J, editors. The NCBI Handbook [Internet]. Bethesda (MD): National Center for Biotechnology Information (US); 2002-. Chapter 16. Available from: http://www.ncbi.nlm.nih.gov/books/NBK21097/
- The Pfam protein families database in 2019: Nucleic Acids Research (2019) doi: 10.1093/nar/gky995)
L319 and throughout: the SI symbol for second is s, not sec. Please correct. In this line, please include rpm
throughout the manuscript: please unify the symbol for liter - capital "L" or small "l"? (e.g. Figure 2a - capital L, caption and text small l). Please make sure that you use en dashes, em dashes, and hyphens correctly and consistently.
L294: please define the crosslinker. Did you use TEMED and APS? Please clarify.
Your reply: "In general, endolysins have a favorable safety profile. Safety studies have been done on many endolysins in cell culture, in animal safety/toxicity studies, and several companies are developing endolysins that
have been through Phase I and Phase II human clinical trials. However, this is the first manuscript describing LysBC17 and it has not been tested on human cells at this time."
Comment: I would recommend the authors to add a sentence or a few words to the manuscript (L170), summarizing their reply. suggestion: "..human body temperature is 37°C and endolysins in general are not toxic to animal cells (or harmful, or something similar - your choice)..."
L253-273: PCR cycling conditions still missing. How long was the amplicon? Please specify in the manuscript.
Your reply: "Yes, the cells were sonicated for 20 min, but at a 25% duty cycle time, which means the total sonication time was 5 minutes. The cycle time information has been added to the text."
Comment: please clarify the regimen and the frequency/intensity in the manuscript.
Author Response
Thank you for corrections, clarifications, and explanations. I can endorse the manuscript for publication after the
authors address the following minor remarks:
L34: producing pneumonia ‐ I was not clear in my previous review report. Please replace the word "producing"
with "causing"
Corrected.
L77‐78 (phylogenetic support): I had in mind a phylogenetic tree. We can also survive without this.
The take home message of this section is that the gene is not associated with prophage elements nor is it
ubiquitously present in all Bacillus genomes. We specify that homologs, which we defined as greater than 80%
coverage and above 85% identity, were only found in 22 out of 147 sequenced Bacillus genomes. To do a
detailed phylogenetic tree of these 22 genes showing >85% identity is a bit tangential to aim of the manuscript.
Thus, given the current length of the manuscript and supplement, we have elected to not include a phylogenetic
tree, but will do so if requested by the editor.
Throughout: Please cite NCBI, Pfam, and ALL other online tools/software/platforms correctly. Examples:
All updated. Note, NCBI states the preferred citation is a 2018 manuscript that lists all resources at the site
rather than the older citation indicated by the reviewer. Additionally, there is no citation for BOXSHADE. The
FAQs at the server state:
“Q: How can I cite BOXSHADE / Is there a publication on BOXSHADE ?
There is no publication on BOXSHADE and none is planned. Most people just use it for figures in publications and
don't mention anything, this is ok for us. If you really feel like mentioning BOXSHADE, you could either
acknowledge it in the figure legend or in the Mat&Meth part on sequence analysis.”
L319 and throughout: the SI symbol for second is s, not sec. Please correct.
“Sec” has been corrected to “s”.
L319 In this line, please include rpm.
The reviewer wants the rpms listed for the shake function of the SpectraMax 340 plate reader. The shake
function is not programmable. It is an automated function of the instrumentation and no details about the rpms
are provided by the user manual, the technical manual, the instrument specifications, or the knowledge database
at the manufacturer’s website. A phone call to the technical staff at Molecular Devices (1‐800‐635‐5577), the
instrument manufacturer, did provide any answers. Thus, no rpm is listed.
Throughout the manuscript: please unify the symbol for liter ‐ capital "L" or small "l"? (e.g. Figure 2a ‐ capital L,
caption and text small l). Please make sure that you use en dashes, em dashes, and hyphens correctly and
consistently.
Corrected. All changed to “L”.
L294: please define the crosslinker. Did you use TEMED and APS? Please clarify.
Information added.
Your reply: "In general, endolysins have a favorable safety profile. Safety studies have been done on many
endolysins in cell culture, in animal safety/toxicity studies, and several companies are developing endolysins that
have been through Phase I and Phase II human clinical trials. However, this is the first manuscript describing
LysBC17 and it has not been tested on human cells at this time."
Comment: I would recommend the authors to add a sentence or a few words to the manuscript
(L170), summarizing their reply. suggestion: "..human body temperature is 37°C and endolysins in general are
not toxic to animal cells (or harmful, or something similar ‐ your choice)..."
Text and a reference added.
L253‐273: PCR cycling conditions still missing. How long was the amplicon? Please specify in the manuscript.
Information added.
Your reply: "Yes, the cells were sonicated for 20 min, but at a 25% duty cycle time, which means the total
sonication time was 5 minutes. The cycle time information has been added to the text."
Comment: please clarify the regimen and the frequency/intensity in the manuscript.
Information added.